# Investigation of the Potential Phlorotannins and Mechanism of Six Brown Algae in Treating Type II Diabetes Mellitus Based on Biological Activity, UPLC-QE-MS/MS, and Network Pharmacology

**DOI:** 10.3390/foods12163000

**Published:** 2023-08-09

**Authors:** Jialiang Chen, Zheng Zhou, Ping Li, Shuhong Ye, Wei Li, Ming Li, Lin Zhu, Yan Ding

**Affiliations:** 1School of Food Science and Technology, Dalian Polytechnic University, Dalian 116034, China; cjliang1998@163.com (J.C.); liping547464x@163.com (P.L.); hlcyeshuhong@hotmail.com (S.Y.); wishzzl@163.com (L.Z.); 2National Engineering Research Center of Seafood, School of Food Science and Technology, Dalian Polytechnic University, Dalian 116034, China; zhouzheng971201@163.com; 3Korean Medicine (KM) Application Center, Korea Institute of Oriental Medicine, Daegu 41062, Republic of Korea; liwei1986@kiom.re.kr; 4College of Basic Medical Science, Dalian Medical University, Dalian 116044, China; vivianmarat@163.com

**Keywords:** phlorotannins, T2DM, UPLC-QE-MS/MS, network pharmacology

## Abstract

Type 2 diabetes mellitus (T2DM) has developed into an important health concern worldwide. The discovery of phlorotannins and their efficacy in the treatment of T2DM has become a hotspot for research in various fields. In this study, the potential phlorotannins and mechanism of six brown algae against T2DM were in-depth investigated using biological activity assays, LC-MS, and network pharmacology. First, the ethyl acetate fraction (EA frac.) showed high polyphenolic content and possessed significantly antioxidant and enzyme inhibitory abilities. Further, a total of fifty-nine peaks were obtained from six EA fracs. via UPLC-QE-MS/MS analysis, and fifteen of them were identified as phlorotannins and their isomers or derivatives. In detail, the chemical structures of six phlorotannins were inferred as dibenzodioxine-1,3,6,8-tetraol, bifuhalol, dioxinodehydroeckol, eckol, fucofurodiphlorethol, and fucotriphlorethol; three phlorotannin isomers were deduced to be fucophlorethol, trifucol, triphlorethol A, or triphlorethol B; and the phlorotannin derivative of *m/z* 263 was determined to be dibenzodioxine-1,2,3,6,8-pentanol or dibenzodioxine-1,2,4,5,7-pentanol. Moreover, 43 T2DM-related targets acted on by these chemicals were identified, and the function of phlorotannin to prevent and treat T2DM was elucidated in a holistic way based on the established compound-target-disease network, and GO function and KEGG pathway enrichment analysis.

## 1. Introduction

As the most common form of diabetes, type 2 diabetes mellitus (T2DM) has become an increasingly serious global health problem [1]. T2DM, also known as adult-onset diabetes, is characterized by a combination of multiple factors, including resistance to insulin action and insufficient compensatory insulin secretion response [2]. This situation will lead to the long-term and gradual development of hyperglycemia. Although it usually does not cause obvious symptoms, it is enough to cause pathological and functional changes in various target tissues of the body [3]. The potential hyperglycemia in diabetes patients will lead to glycotoxicity, which will further induce the occurrence of chronic complications in multiple systems and organs (such as microvascular and macrovascular diseases) [4]. In addition, continuous hyperglycemia also leads to excessive free radical release [5]. Oxidative stress, advanced glycation end products (AGEs), and inflammation are inextricably intertwined in the context of diabetes and related complications [6]. Oxidative stress induced by hyperglycemia triggers cells of the innate immune system to release proinflammatory cytokines, which leads to excessive production of free radicals by activating macrophages [7].

At present, the treatment of T2DM mainly includes insulin and various oral antidiabetic drugs, such as α-glucosidase inhibitors, rosiglitazone, sulfonylurea, metformin, and thiazolidinedione [8,9]. However, these drugs have significant side effects, such as hypoglycemia, weight gain, and increased gastrointestinal problems [10]. Due to their wide range of plant sources and low toxicity, natural products have attracted more and more people’s interest in alternative therapy and natural medicine for diabetes [11,12].

Phenolic compounds, the most studied phytochemicals, are ubiquitous and abundant in all plant-based diets. It is well known that marine algae contain rich bioactive compounds, which have great potential for pharmaceutical foods and biomedicine [13]. As polyphenols unique to brown algae, phlorotannins are different from terrestrial plant polyphenols consisting of gallic acid and flavonoid polymers. Phlorotannins are polymeric derivatives of phloroglucinol units (1,3,5-trihydroxybenzene) [14]. Influenced by the connection mode between the phloroglucinol units and the distribution of hydroxyl groups, phlorotannins can be classified into six major subtypes: fucols, phlorethols, fucophlorethols, eckols, carmalols, and fuhalols [15]. Phlorotannins have a wide range of health benefits, such as anti-inflammatory, anticancer, antioxidation, and antibacterial, which has aroused research interest. Some studies have even reported that there is great potential for the use of phlorotannins as a functional ingredient in the development of new and improved food products, mainly in the areas of dairy and seafood products [13]. Moreover, many studies have pointed out the antioxidant activity [16] and antidiabetes effect of phlorotannins [17].

Therefore, we aimed at exploring the in vitro effect of phlorotannin-targeted extract from six brown algae on the metabolic changes of diabetes, that is, its regulation is involved in carbohydrate hydrolase (α-glucosidase and α-amylase) and antioxidant activity; mass spectrometry was used to identify the phlorotannins and determine the correlation between the polyphenol components and the anti-T2DM activity, predicting the mechanism of phlorotannins against T2DM using a network pharmacology approach. The above research provides a strong basis for the application of phlorotannins (Figure 1).

## 2. Materials and Methods

### 2.1. Chemicals and Reagents

Ethanol, n-hexane, and ethyl acetate were purchased from Guangdong Guanghua Sci-Tech Co., Ltd. (Shantou, China). Potassium dihydrogen phosphate, dipotassium hydrogenphosphate, ascorbic acid (≥99.0%), trichloroacetic acid, Iron(III) chloride (FeCl_3_), starch, and sodium hydroxide were purchased from Damao Chemical Reagent Factory (Tianjin, China). Phloroglucinol (≥99.0%), sodium carbonate, diammonium salt (ABTS), α-glucosidase, α-amylase, 3,5-dinitro salicylic acid, and potassium sodium tartrate were purchased from Shanghai Aladdin Biochemical Technology Co., Ltd. (Shanghai, China). 4-nitrophenyl-α-D-glucopyranoside (PNPG), acarbose, 2,2-diphenyl-1-picrylhydrazyl radical (DPPH), and Folin-Ciocalteu reagent were purchased from Beijing Solarbio Technology Co., Ltd. (Beijing China). Potassium ferricyanide and potassium persulfate were purchased from Xilong Science Co., Ltd. (Shantou, China). Methanol was chromatographic grade and purchased from Concord Technology (Tianjin, China).

### 2.2. Databases and Software

The following databases and software were used: SwissTargetPrediction database (https://www.swisstargetprediction.ch (accessed on 12 April 2023)), the human gene integrator (GeneCards, https://www.genecards.org (accessed on 13 April 2023)), Metascape database (https://metascape.org/ (accessed on 16 April 2023)), Cytoscape version 3.8.2, online interactive tool for Venn (http://jvenn.toulouse.inra.fr/ (accessed on 20 April 2023)), ChemOffice version 2019.

### 2.3. Brown Algae Samples

*Undaria pinnatifida* (*U. pinnatifida*), *Sargassum pallidum* (*S. pallidum*), *Scytosiphon lomentarius* (*S. lomentarius*), *Sargassum hemiphyllum* (*S. hemiphyllum*), *Sargassum fusiforme* (*S. fusiforme*), and *Laminaria japonica* (*L. japonica*) were purchased from Dalian local supermarkets (Liaoning, China). After removing impurities, the algae samples were dried at 50 °C and ground into powder (particle size < 0.6 mm). The obtained powder samples were vacuum-packed and stored below −30 °C until further experiments.

### 2.4. Sample Preparation and Extraction

The extraction of phlorotannin compounds was carried out using 80% ethanol solution based on the method of Park et al. [18]. For this purpose, 25 g dried algal powder was dispersed in 500 mL 80% ethanol solution (*v/v*) and extracted for 90 min. The ethanol extract (EE) was filtered through a suction filter to remove impurities and concentrated to a volume of about 50 mL in a rotary evaporator. The concentrated extract was extracted several times using n-hexane (1:1, *v/v*) until n-hexane appeared colorless. The aqueous phase was then subjected to further liquid–liquid extraction using ethyl acetate (1:1, *v/v*). Solvents in ethanol extract, n-hexane (HEX), ethyl acetate (EA), and aqueous (AQ) fractions were evaporated and stored at −20 °C until analysis [19].

### 2.5. Determination of Total Phenolic Content

The total phenolic content of the samples was determined using the Folin-Ciocalteu method [20]. A total of 10 µL phlorotannins extract and 30 µL of Folin-Ciocalteu reagent were added to a 96-well plate and mixed thoroughly. After sitting for 3 min, 80 µL of 0.1 g/mL sodium carbonate solution was added to the 96-well plate. After complete mixing, deionized water was added until the volume reached 200 µL. After 25 min of reaction in the dark at 30 °C, the absorbance was measured at 760 nm. The measured values were compared with the standard curve prepared with phloroglucinol solution. The total phenolic content was expressed as the content of phloroglucinol per gram of dried extract (DW) (y = 1.4717x + 0.0171, R^2^ = 0.9987).

### 2.6. Antioxidant Activities

#### 2.6.1. DPPH· Scavenging Test

DPPH scavenging activity was determined according to the procedure described by Catarino et al. [21]. A total of 100 μL sample solution and 100 μL of DPPH solution were added sequentially to a 96-well plate and then mixed thoroughly. The reaction was carried out for 30 min at 25 °C under the conditions of a dark environment and the absorbance values were measured at 517 nm. In the blank group, ultrapure water was used instead of the sample and the absorbance value was Ac. To eliminate the absorbance value of the sample itself it was also necessary to set up a sample blank group with methanol instead of DPPH solution. The DPPH-clearance of the experimental samples was calculated using Vc as a control.
(1)$$\mathrm{Inhibitory}\;\mathrm{rate}\;(\%)=\frac{Ac-As}{Ac}\times 100$$
here *Ac* and *As* represent the strength of the control and inhibitor, respectively.

#### 2.6.2. ABTS· Scavenging Test

ABTS scavenging activity was determined according to the procedure described by Nickavar et al. [22]. The preparation of ABTS solution was first performed: 38.4 mg ABTS and 13.5 mg potassium persulfate were dissolved into a constant volume of 10 mL each, mixed, and stored away from light for 12 h. The ABTS solution was diluted to 45 times with PBS (pH = 6.8) buffer solution. A total of 25 μL sample solution and 175 μL ABTS solution were added to a 96-well plate, the reaction was carried out at room temperature in the dark for 30 min, and the absorbance value was measured at 734 nm. In the blank group, ultrapure water was used instead of the sample and the absorbance value was Ac. To eliminate the absorbance value of the sample itself it was also necessary to set up a sample blank group with PBS instead of ABTS solution. The ABTS radical scavenging rate of the samples was calculated using Vc as the control.

#### 2.6.3. Iron Ion Reduction Force Experiment

Some modifications were made to the method for assessing the ferric-ion-reducing antioxidant power according to Chew’s description [23]. A total of 400 μL PBS buffer (pH 6.8), 400 μL potassium ferricyanide (1%), and 400 μL sample solution were mixed and reacted at 50 °C for 20 min. After cooling to room temperature, 400 μL trichloroacetic acid (10%) was added to the mixed solution. After the reaction, centrifuge at 3000 r/min for 10 min, take 50 μL supernatant and mix it with 100 μL distilled water and 50 μL ferric chloride. The mixture was incubated for another 10 min at room temperature to develop the color and the absorbance was measured at 700 nm. The standard curve was plotted with vitamin C solution (y = 3.7444x + 0.0818, R^2^ = 0.9940). The results are shown as Vc μg/g DW.

### 2.7. Enzyme Inhibition Assay

#### 2.7.1. α-Glucosidase Inhibition Assay

Appropriate improvements were made to the existing α-glucosidase assay [18]. A total of 100 μL sample solution was added to a 96-well plate, followed by 50 μL α-glucosidase solution (1 unit/mL, pH 6.8). Finally, 50 μL PNPG solution (0.6 mg/mL) was added to the 96-well plate. The reaction was carried out under lightproof conditions for 30 min, and then the absorbance of the sample was measured at 405 nm. The inhibitory capacity of acarbose was used as a reference.

#### 2.7.2. α-Amylase Assay

The ability of phlorotannins extract to modulate α-amylase activity was assessed using a colorimetric method with DNS acid [24]. A total of 40 μL sample solution and 40 μL starch solution (1%) were mixed and incubated for 10 min at room temperature. Then, 40 μL α-amylase (0.5 mg/mL) was added and incubated for another 10 minutes. Finally, 80 μL DNS acid (1% in 20% NaOH and 30% KNaC_4_H_4_O_6_·4H_2_O) was added to the mixed solution and heated at 100 °C for 5 min. The absorbance of the reaction products was measured at 540 nm. The pharmacological inhibitor acarbose was selected as a positive control.

### 2.8. Characterization of Phlorotannins

The mass spectrometry method of Shen et al. has been properly modified [25]. Analyses were performed in a Q-Exactive HF-X Hybrid Quadrupole-Orbitrap (Q-Exactive HF-X) mass spectrometer with electrospray ionization (ESI) using a Dionex UltiMate 3000 system (Dionex Softron GmbH, Germany) with an Acquity UPLC BEH HILIC column (2.1 × 150 mm, particle size 1.7 µm; Waters Corporation, Milford, MA, USA). The final concentration of polyphenolic extracts of six kinds of brown macroalgae was 2 mg/mL, and the dissolving solvent was methanol. Run conditions were as follows: 25 °C column temperature; 0.3 mL/min final flow rate; 2 µL injection volume; and 280nm detection wavelength. Mobile phase A was 0.1% formic acid in water. Mobile phase B was acetonitrile containing 0.1% formic acid. Polyphenols were separated by the following gradient elution: 0–10 min, 5–40% B; 10–12 min, 40–95% B; 12–13 min, 95% B; and 13–15 min, 95–5% B. Spectra acquisition was performed in positive and negative ionization modes, and spectra were acquired over a mass range of *m/z* 60–900. In both modes, the sheath gas flow rate was 60%, the aux gas flow rate was 20%, and the sweep gas flow rate was 1%. The spray voltage was at 3.6 kV, capillary temperature was 380 °C, and aux gas heater temperature was 370 °C.

### 2.9. Network Pharmacological Analysis

#### 2.9.1. Potential Target Prediction and Screening

The screened compounds were saved as SDF files and loaded into the SwissTargetPrediction database for target prediction. The target attribute was set to “Homo sapiens” and target proteins with a probability greater than 0 were listed [26]. The search term “Type 2 diabetes mellitus” was used to search for disease-related targets in the GeneCards database [27]. Phlorotannins and T2DM-related targets were normalized to Uniprot ID in the Uniprot protein database [28], and phlorotannins and disease-related targets were entered separately into the Venn online mapping tool [29], where intersecting targets were considered as potential targets for phlorotannins against T2DM.

#### 2.9.2. GO Function and KEGG Pathway Enrichment Analysis

The Metascape database was used to enrich the Gene Ontology (GO) function and the Kyoto Encyclopedia of Genes and Genomes (KEGG) pathways of the intersecting targets. The GO function and KEGG pathway enrichment analysis can provide a more systematic and comprehensive understanding of the targeting mechanisms between targets and diseases [30]. For this purpose, the intersecting targets were uploaded to the Metascape database and the species entered/analyzed was set to “Homo sapiens”. *p*-values, minimum counts, and enrichment factors were set to 0.01, 3, and 1.5, respectively, and the top 20 were selected for analysis. The Bonferroni method was used to correct for major biological processes and the KEGG pathway, and the enrichment score (-log10(Q-value)) was used to estimate correlation. The Origin software was used to plot the color bubbles to be analyzed, with the color and size of the circles in the graph representing the Q-value and target counts, respectively.

#### 2.9.3. Network Construction of Compound-Target-Disease

To analyze the association between active compounds, targets, and diseases, a Compound-Target-Disease (C-T-D) network was constructed using Cytoscape 3.8.2 software [31]. In the network, the size of the nodes is proportional to their degree, and differences in color and shape between the nodes represent the properties of the active compound, target, and disease, respectively.

### 2.10. Statistical Analysis

All experiments were repeated at least three times. Final results were presented as mean ± SD. Statistical determination was computed using a Student’s t-test or one-way analysis of variance using SPSS, version 20 (IBM Corp., Armonk, NY, USA). *p* < 0.05 was considered statistically significant.

## 3. Results and Discussion

### 3.1. Evaluation of Total Phenolic Content

In this study, the total phenolic content (TPC) of six brown algae extracts showed significant difference between species and the extraction fraction (*p* < 0.05), ranging from 0.113 to 6.315 (g PGE/100 g DW) (Table 1). Among the analyzed samples, the EA of *S. hemiphyllum* has the highest TPC (6.315 g PGE/100 g DW), while the AQ of *U. pinnatifida* has the lowest TPC (0.113 g PGE/100 g DW). The polyphenol compounds of *U. pinnatifida* mainly existed in the EA frac., and TPC of the EA frac. was 13.81 times (1.56 g PGE/100 g DW) its water phase. In the different extraction fraction, the phlorotannins mainly exist in the EA frac. However, the enrichment value of *L. japonica*-EA is 0.2. This might be explained by the fact that there are few phenolic compounds in *L. japonica*, or that the Folin-Ciocalteu method not only measures the reduction ability of polyphenols but also measures the reduction ability of ascorbic acid, aromatic amine, sugar, and other reducing substances [32]. Polar and nonpolar compounds are not completely insoluble in the extraction process, which may lead to a part of polyphenol compounds being taken away by the water phase. The analysis results of Dong et al. show that the yield of phenolic compounds in *Undaria pinatifida* is 10.7 ± 0.2 mg gallic acid equivalent/g dry weight (GAE/g DW) of the sample [33]. A similar total phenol content was reported in other brown algae [34,35].

### 3.2. Antioxidant Activities

The antioxidant activity of different solvent-extracted phlorotannins from six brown algae was evaluated using DPPH, ABTS, and FRAP radical scavenging assays, and the relationship between activity and TPC was analyzed [19]. Significant differences in antioxidant activity were found in twenty-four fractions from six brown algae (Table 2). *S. pallidum*-EA showed the highest DPPH activity (IC_50_ 0.33 ± 0.03 mg/mL), followed by *S. hemiphyllum*-EA (IC_50_ 0.35 ± 0.01 mg/mL), and *U. pinnatifida*-AQ showed the lowest DPPH activity (IC_50_ 60.37 ± 3.04 mg/mL). Similar trends were observed for ABTS radical scavenging capacity and FRAP activity, with *S. hemiphyllum*-EA having the highest ABTS radical scavenging capacity (IC_50_ 0.02 ± 0.00 mg/mL), followed by *S. pallidum*-EA (IC_50_ 0.11 ± 0.01 mg/mL). The EA fracs. of *S. hemiphyllum* and *S. pallidum* have stronger ABTS inhibition than ascorbic acid (IC_50_ 0.27 ± 0.00 mg/mL). When measured using FRAP, the activity of *S. hemiphyllum*-EA and *S. pallidum*-EA was 53.74 ± 5.52 and 22.94 ± 0.81 mg Vc/g DW, respectively. The results obtained presented similar results with the antioxidant potential of different extract fractions of *Sargassum vulgare*. Among them, considerable antioxidant potential was shown in the EtOAc fraction, which was more promising than the crude extract, Hex, and AQ [19].

A strong association or relationship between antioxidant activity (DPPH, ABTS, and FRAP) and TPC has been observed relative to TPC. The components with higher content of phlorotannins have stronger antioxidant capacity. This is consistent with previous studies that correlated high antioxidant activity with high levels of polyphenols [36,37]. In general, EA frac. shows stronger antioxidant potential than the crude extracts Hex and AQ, which may be related to its higher polyphenol content in brown algae [38]. In addition, the geometrical arrangement of the polyphenol structure and the location of free radicals may also affect the antioxidant activity [39].

### 3.3. Enzyme Inhibition Assay

The control of diabetes usually involves regulating key carbohydrate metabolic enzyme (α-glucosidase and α-amylase) activity to stabilize blood glucose levels. In this study, there was a similar trend between the inhibition of carbohydrate hydrolase and antioxidant activity. Although to different extents, the solvent fractions of six brown algae were able to inhibit enzyme activity (Table 2). *S. hemiphyllum*-EA had the highest inhibition of enzyme activity (α-glucosidase: IC_50_ 0.02 ± 0.00 mg/mL, α-amylase: IC_50_ 0.02 ± 0.00 mg/mL), followed by *S. pallidum*-EA (α-glucosidase: IC_50_ 0.05 ± 0.01 mg/mL, α-amylase: IC_50_ 0.51 ± 0.01 mg/mL). The EA frac. Of *S. hemiphyllum* and *S. pallidum* have a significantly stronger inhibitory effect on α-glucosidase and α-amylase than the pharmacological inhibitor acarbose (α-glucosidase: IC_50_ 0.63 ± 0.06 mg/mL, α-amylase: IC_50_ 2.51 ± 0.11 mg/mL). When the concentration of *L. trabeculate* extract reaches 10 mg/mL, α-glucosidase can be completely inhibited [40]. The IC_50_ of acetone extract of *S. pallidum* against α-glucosidase was 270.7 μg/mL [41]. Chen et al. confirmed that the EA frac. Of six brown algae had higher α-glucosidase inhibitory activity than acarbose [42]. Ismail et al. found a correlation between the TPC of beech extracts and the carbohydrate metabolic enzymes activity [43].

### 3.4. UPLC Analysis of Different Components

The study of phlorotannin constituents in six brown algae is important for a better understanding of their activities and further development of their applications. The EA frac. Showed higher TPC, antioxidant activity, and enzyme inhibition ability, thus it was further analyzed by Q-Exactive HF-X mass spectrometry. Figure 2 shows the base peak ion-chromatograms (BPC) of ethyl acetate fraction at 280nm.

In general, a total of 59 phlorotannins and derivatives with molecular weights of 2–5 phloroglucinol units were detected in the EA frac. of six brown algae. Retention times (Rt), precursor ions, product ions, tentative assignment, and species for each identified phlorotannins are presented in Table 3. There were significant differences in the distribution of compounds between samples, with a total of 26 peaks for phlorotannins and their derivatives obtained from the *S. fusiforme* sample, followed by *S. lomentarius* (24), while the least number of phlorotannins and their derivatives were identified in *U. pinnatifida* (4). Of the 59 peaks screened, peaks 3, 8, 45, and 47 were present in three or more brown algae simultaneously.

In this study, 15 compounds were identified as phlorotannins and their isomers or derivatives. A total of six compounds were properly characterized according to the MS ^2^ spectral and the data in the literature (Figure 3). These were as follows: dibenzodioxine-1,3,6,8-tetraol ([M-H]^−^ at *m*/*z* 247), bifuhalol ([M-H]^−^ at *m*/*z* 265), dioxinodehydroeckol ([M-H]^−^ at *m*/*z* 369), eckol ([M-H]^−^ at *m*/*z* 371), fucofurodiphlorethol ([M-H]^−^ at *m*/*z* 479), and fucotriphlorethol ([M-H]^−^ at *m*/*z* 621) [10,45,46].

Peak 8 was identified as dibenzodioxine-1,3,6,8-tetraol ([M-H]^−^ at *m*/*z* 247), a phlorotannin present in four brown algae, *S. pallidum*, *L. japonica*, *U. pinnatifida*, and *S. fusiforme*. The main product ions of this compound MS/MS were 228.96, 202.92, and 166.96, corresponding to the [M-H-18]^−^, [M-H-44]^−^, and [M-H-44-18-18]^−^ groups, respectively [10,19,44]. The peak 8 is a dimer of phloroglucinol, which may be precursors of eckol-type phlorotannins. In addition, the compound was found for the first time in these four brown algae. Bifuhalol ([M-H]^−^ at *m*/*z* 265) has fragment ions at *m*/*z* 247.09 ([M-H-18]^−^) and *m*/*z* 221.12 ([M-H-18-18]^−^). In addition, fragment ions at *m*/*z* 123.04 are considered to be caused by the loss of a pyrogallol unit and water molecules [47]. Peak 51 was present only in the sea jute thread sample and was structurally identified as eckol.

It is difficult to determine the correct spatial structure of these phlorotannins through LC-MS because the connection position of phloroglucinol unit cannot be determined accurately. The phlorotannins structure shown in Figure 4 represents the compounds that may exist in different isomers. The mass spectrum data of peaks 11, 26, and 36 show that they have the same *m*/*z* (373), and they are considered as trimers of phloroglucinol. However, due to the different linking methods, peaks 11, 26, and 36 can be considered as isomeric phlorotannins. Among them, 247.02 and 229.01 in the mass spectrum of peak 11 are considered to have removed one phloroglucinol unit and water. The values 329.17 and 305.18 in peak 26 and 36 are considered to have removed one molecule of carboxylic acid and two carbon atoms. According to the mass spectrum information in the literature, the peaks 11, 26, and 36 may be fucophlorethol, trifucol, triphlorethol A, and triphlorethol B [10].

The formation of phlorotannin derivatives may be caused by an increase or decrease in hydroxyl groups or water molecules on phlorotannins or by the conversion of fuhalol-type to eckol-type. Based on the information of ion fragments of derivatives of phlorotannin in six brown algae, we deduced the structures of the obtained compounds. The compounds with a mass to nucleus ratio of 263 are thought to have been formed in two ways: the polymerization of a ring between two benzene rings in bifuhalol and the addition of a hydroxyl bond to the benzene ring in dioxinodehydroeckol (Figure 5). The derivatives of phlorotannin with a mass-to-nucleus ratio of 263 are named dibenzodioxine-1,2,3,6,8-pentanol or dibenzodioxine-1,2,4,5,7-pentanol according to the nomenclature of phlorotannin. [M-H]^−^ at *m*/*z* 387 is considered to be a phloroglucinol unit connected by a C-C bond on the basis of *m*/*z* 263. The compound with *m*/*z* 385 is considered to have added a hydroxyl bond on the basis of dioxinodehydroeckol (*m*/*z* 369); [M-H]^−^ at *m*/*z* 711 is believed to have removed a hydroxyl bond on the basis of dieckol (*m*/*z* 742); and [M-H]^−^ at *m*/*z* 403 and 421 is considered to have added two and three hydroxyl bonds on the basis of triphlorethol (*m*/*z* 247), respectively [44].

Notably, even though the structure of all phlorotannins has not been clarified, the structure of six derivatives of phlorotannin has been further deduced based on their MS ^2^ spectra. This is mainly because its derivatives show a fragmentation pattern similar to that of phlorotannins, or produce ions indicating one or more phloroglucinol units and characteristic products, as well as common water loss or cross ring breakage [44,48].

### 3.5. Network Pharmacological Analysis

#### 3.5.1. Potential Target Prediction and Screening

The structures of the identified phlorotannin were drawn by ChemBioDraw software and uploaded to the SwissTargetPrediction database to predict a total of 48 relevant targets (Probability > 0). Using the keyword “Type 2 diabetes mellitus”, 13594 T2DM-related targets were retrieved from the GeneCards database. Using the Venn online tool to map the active ingredients to T2DM-related targets, a total of 43 intersecting targets were obtained (Figure 6), which are potential targets of phlorotannin for the prevention and treatment of T2DM.

#### 3.5.2. GO Function and KEGG Pathway Enrichment Analysis

Using the Metascape database, the GO function and KEGG pathway were analyzed. A total of 600 GO items were associated with the effect of phlorotannins against T2DM, including 490 biological process (BP) items, 48 cellular component (CC) items, and 62 molecular function (MF) items. Each item was ranked according to its q-value and the top ranked ones were presented visually, as shown in Figure 7a. The results of the enrichment analysis showed that the target proteins were mainly involved in biological processes such as the positive regulation of phosphorylation, positive regulation of transferase activity, protein phosphorylation, regulation of kinase activity, and positive regulation of kinase activity. Phlorotannin regulatory targets were distributed in cellular components such as the membrane raft, membrane microdomain, cell body, growth cone, site of polarized growth, and axon. The molecular functions of phlorotannin-related targets are mainly related to carbonate dehydratase activity, hydro-lyase activity, carbon-oxygen lyase activity, lyase activity, protein kinase activity, and phosphotransferase activity.

A total of 98 signaling pathways were obtained from the KEGG pathway enrichment analysis, and the top 20 pathways were selected for presentation based on q-value ranking, as shown in Figure 7b. The top KEGG pathways include the EGFR tyrosine kinase inhibitor resistance, pathways in cancer, endocrine resistance, PI3K-Akt signaling pathway, fluid shear stress and atherosclerosis, and Rap1 signaling pathway.

Inhibition of epidermal growth factor receptor (EGFR) activity protects diabetic patients from DN [49]. PI3K/AKT signaling pathway is the major insulin regulatory pathway in the liver and is responsible for the regulation of glucose transport, β-cell secretion, and insulin gene transcription. Transduction of this pathway is triggered when insulin binds to specific receptors on the surface of hepatocyte membranes, promoting phosphorylation of downstream AKT. Activation of AKT promotes glucose transporter protein 2 (GLUT2) transport, thereby regulating glucose metabolism. Studies have shown that phlorotannins can play a potential role in combating T2DM at a systemic level by modulating pathways associated with T2DM and its complications.

#### 3.5.3. Compound-Target-Disease Network Analysis

In the C-T-D network, yellow squares represent targets, red circles represent diseases, and green hexagons represent phlorotannins (Figure 8). The C-T-D network contains a total of 55 nodes and 151 edges, and almost all target proteins have an interaction relationship with each other compound. This result suggests that most of the targets are regulated by multiple components and that the same substance also regulates several target proteins simultaneously. BACE1, MAOB, AKT1, ESR1, ESR2, IGF1R, NOX4, and SRC in the C-T-D network are closely associated with T2DM. AKT1 is involved in a variety of cellular processes, including apoptosis and glucose metabolism, which are associated with diabetes, tumors, rheumatoid arthritis, and many other diseases [50]. Estrogen receptor 1 (ESR1), a transcription factor, promotes the expression of the glucose transporter type 4 (SLC2A4) gene, which increases GLUT4 levels in tissues, thereby reducing serum glucose levels [51]. Inducible NO synthase inhibitor (L-NIL) has a significant ameliorative effect in tetraoxopyrimidine diabetic female rats and can cause a significant increase in the expression of ESR2 in the aortic wall of rats. It is possible that ESR2 is involved in the expression of NO synthase (iNOS), which exacerbates tetraoxopyrimidine-induced type 1 diabetes [52]. Insulin-like growth factor (IGF)-1 is an important cellular regulator that increases IGF-1 (IGF-1) secretion and upregulates IGF-1 receptor (IGF1R) in a high-glucose environment, which in turn exacerbates renal damage [53]. NOX4, the most widely studied NOX isoform in DN, is a major source of ROS production in the kidney [54]. Prolonged stimulation with high glucose allows for induced activation of NOX oxidase, which in turn causes the increased expression of NOX4 in thylakoid cells and results in elevated ROS levels [55].

## 4. Conclusions

The present study was designed to provide a scientific basis for the analysis of potentially active phlorotannins against T2DM via component isolation, bioactivity determination, UPLC-QE-MS/MS, and network pharmacology. Antioxidant activity and inhibition of carbohydrate hydrolases were assessed in 24 groups of brown algae samples, and the structures of phlorotannins and their derivatives were deduced via UPLC-QE-MS/MS. Predicting the mechanism of phlorotannins against T2DM using a network pharmacology approach. Polyphenolic compounds and bioactivities differed significantly among the brown algal species evaluated. In addition, the biological activity of polyphenols was closely related to the total phenolic content. In the samples analyzed, phlorotannins were predominantly present in the EA frac., with the highest TPC in the *S. hemiphyllum*-EA. The *S. pallidum*-EA showed the highest DPPH scavenging activity, similar results were shown for other biological activity. A total of six phlorotannins, three phlorotannin isomers, and six phlorotannin derivatives were inferred. A total of six phlorotannins were deduced as dibenzodioxine-1,3,6,8-tetraol, bifuhalol, dioxinodehydroeckol, eckol, fucofurodiphlorethol, and fucotriphlorethol. Dibenzodioxine-1,3,6,8-tetraol was first identified from *S. pallidum*, *L. japonica*, *U. pinnatifida*, and *S. fusiforme*. The phlorotannin isomers correspond to fucophlorethol, trifucol, triphlorethol A, or triphlorethol B. The chemical structure of the phlorotannin derivative of *m/z* 263 was deduced and the two forms present were named dibenzodioxine-1,2,3,6,8-pentanol or dibenzodioxine-1,2,4,5,7-pentanol. A total of 43 T2DM-related targets were obtained via network pharmacological prediction, mainly focusing on EGFR tyrosine kinase inhibitor resistance, endocrine resistance, the PI3K-Akt signaling pathway, and the Rap1 signaling pathway. Phlorotannins were unique among brown algae and showed great physiological activities. Considering the synergistic effects of other active substances, further studies combining multiple analytical methods are needed to reveal the active substances in brown algae more comprehensively and accurately. The in vivo and vitro studies supplemented the targets and mechanisms of the effects of phlorotannins on T2DM to promote the development of related new drugs, which provided a certain theoretical basis for the exploitation of the active value of phlorotannins and maximized the potential of brown alga. This highlights the potential use of brown algae for future commercial applications and their suitability as a model species for further studies of brown algal polyphenols.

## Figures and Tables

**Figure 1 foods-12-03000-f001:**
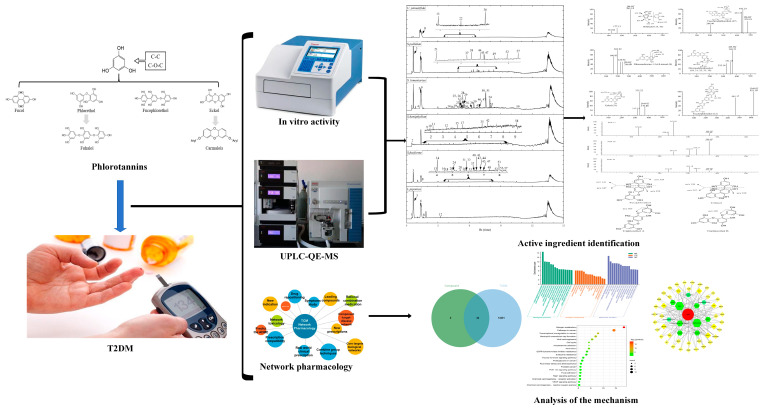
Whole technical route of present research.

**Figure 2 foods-12-03000-f002:**
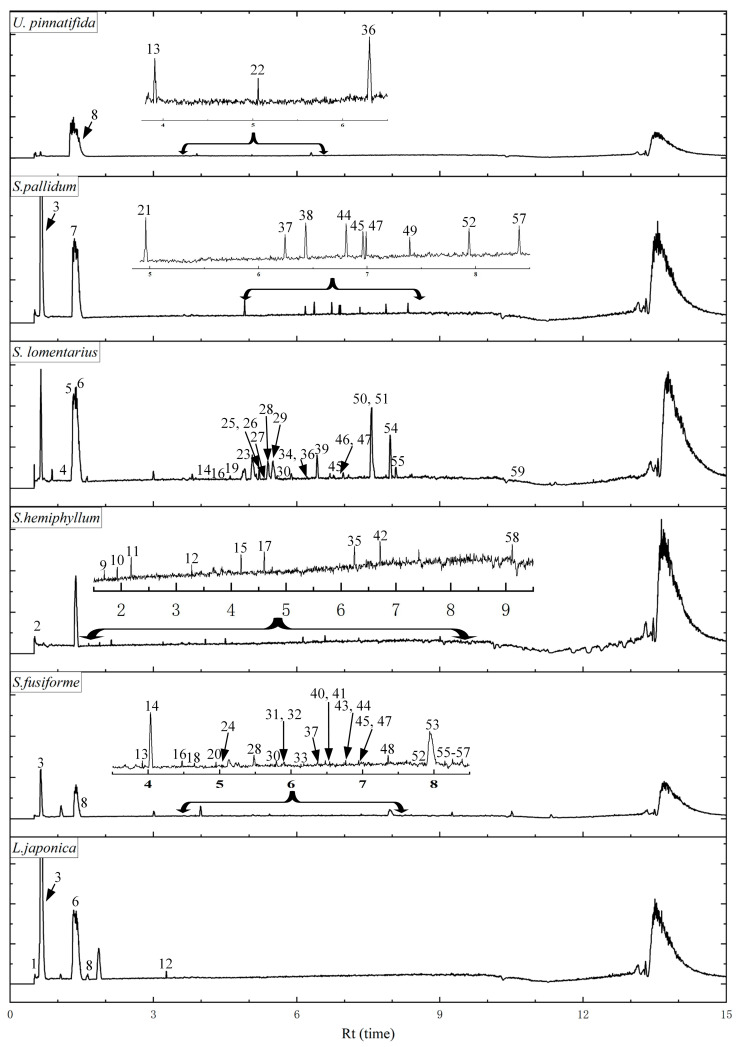
Full scan base peak ion-chromatograms (BPC) of the EA fraction of six brown algae.

**Figure 3 foods-12-03000-f003:**
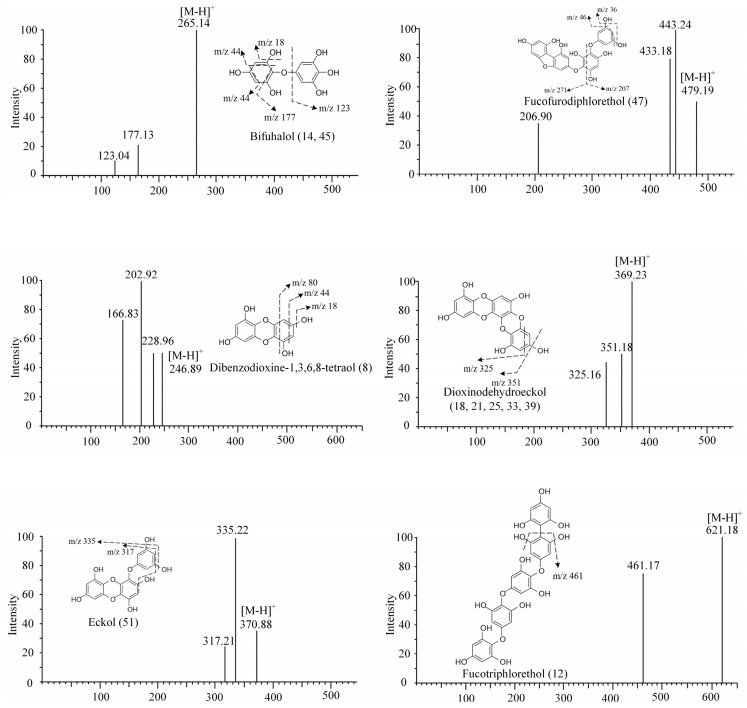
MS/MS fragmentation pattern of identified phlorotannin compounds in ethyl acetate fraction of brown algae.

**Figure 4 foods-12-03000-f004:**
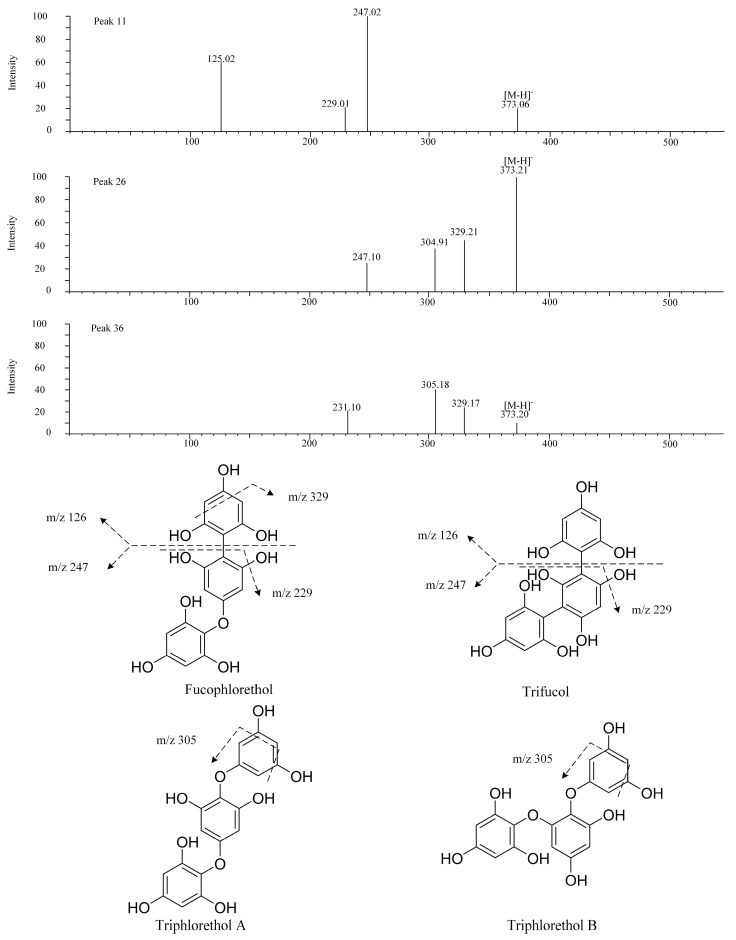
MS/MS fragmentation pattern of identified the isomers of phlorotannin.

**Figure 5 foods-12-03000-f005:**
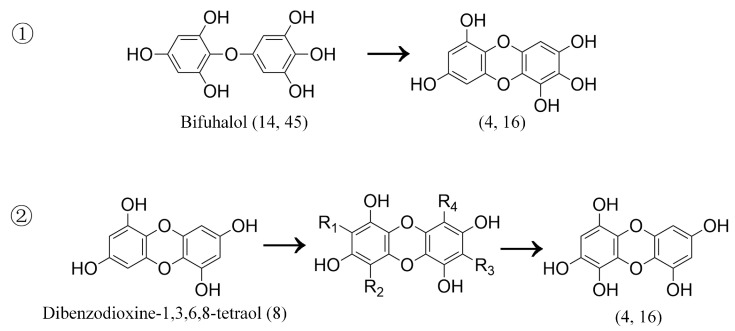
Formation mode of phlorotannin derivative (*m*/*z* 263).

**Figure 6 foods-12-03000-f006:**
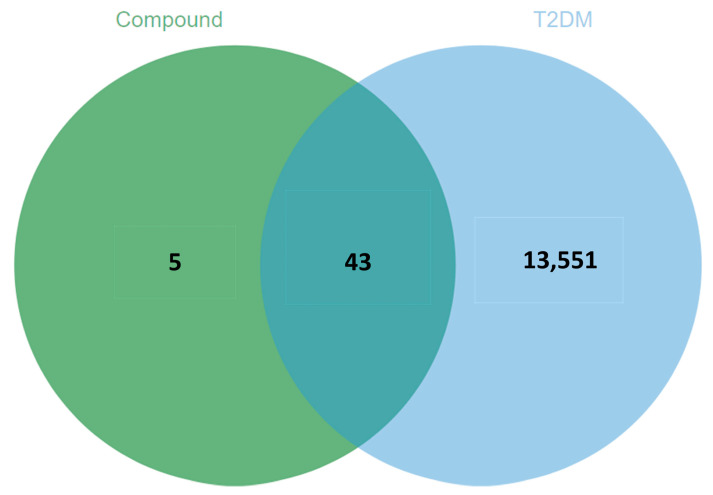
Venn diagram of related targets of active ingredients and T2DM.

**Figure 7 foods-12-03000-f007:**
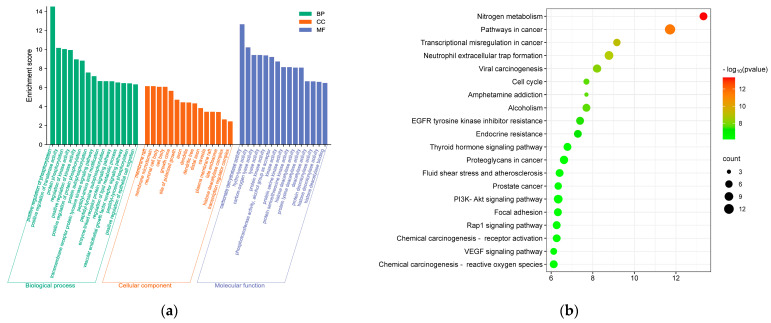
Enrichment analysis of core targets: (**a**) GO function; (**b**) KEGG pathway.

**Figure 8 foods-12-03000-f008:**
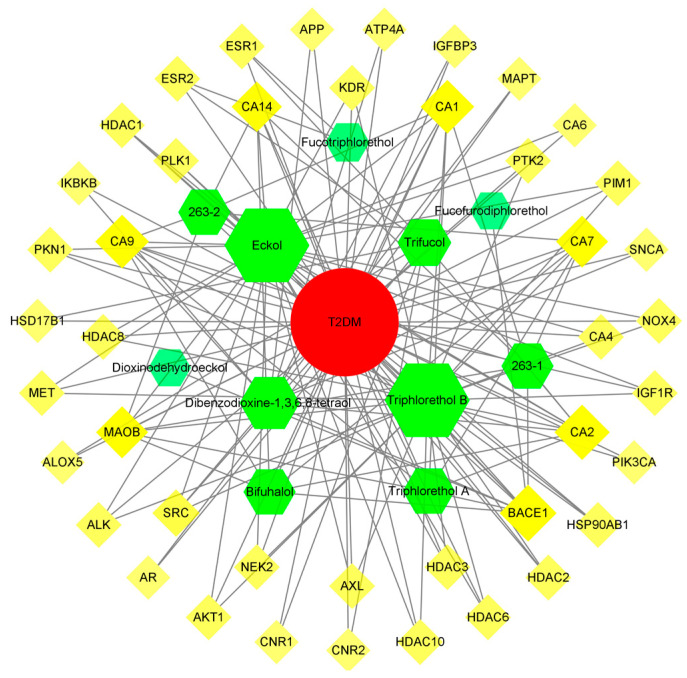
Compound-target-disease network diagram.

**Table 1 foods-12-03000-t001:** Total solids content and total phenolic content of six brown algae crude extract and purified fractions.

No.	Sample		Total Solids (mg)	TPC (g PGE/100 g DW)	Enrichment (Fold)
1	*U. pinnatifida*	EE	4534.8	0.50	/
2		HEX	1164.4	0.64	1.28
3		EA	810.4	1.56	3.12
4		AQ	2560.0	0.11	0.23
5	*S. pallidum*	EE	2956.0	0.74	/
6		HEX	350.3	3.49	4.72
7		EA	206.7	2.34	3.16
8		AQ	2399	0.20	0.06
9	*S. lomentarius*	EE	6948.4	0.30	/
10		HEX	754.9	0.68	2.25
11		EA	326.5	1.60	5.32
12		AQ	5867.0	0.24	0.78
13	*S. hemiphyllum*	EE	1842.4	2.28	/
14		HEX	1021.4	2.85	1.25
15		EA	132.6	6.32	2.78
16		AQ	688.4	0.58	0.25
17	*S. fusiforme*	EE	1359.6	1.12	/
18		HEX	82.8	1.80	1.60
19		EA	114.0	1.90	1.70
20		AQ	1162.8	1.02	0.91
21	*L. japonica*	EE	622.0	3.57	/
22		HEX	246.0	0.75	0.21
23		EA	64.2	0.71	0.20
24		AQ	311.8	4.08	1.14

**Table 2 foods-12-03000-t002:** Activities of antioxidant and enzyme inhibition of six brown algae crude extract and purified fractions.

No.	Sample		DPPH (IC_50_ mg/mL)	ABTS (IC_50_ mg/mL)	FRAP (Vc mg/g DW)	α-Glucosidase (IC_50_ mg/mL)	α-Amylase (IC_50_ mg/mL)
1	*U. pinnatifida*	EE	4.25 ± 0.53	23.08 ± 3.25	14.44 ± 1.04	11.06 ± 0.39	2.44 ± 0.03
2		Hex	1.67 ± 0.15	0.40 ± 0.01	18.53 ± 2.12	0.61 ± 0.08	2.16 ± 0.03
3		EA	2.14 ± 0.03	0.51 ± 0.06	16.22 ± 0.28	0.89 ± 0.10	2.17 ± 0.04
4		AQ	60.37 ± 3.04	13.78 ± 1.27	14.17 ± 0.28	116.90 ± 16.02	8.33 ± 1.81
5	*S. pallidum*	EE	4.43 ± 0.38	2.16 ± 0.08	13.50 ± 0.76	0.65 ± 0.06	3.37 ± 0.26
6		Hex	1.08 ± 0.02	0.69 ± 0.04	18.80 ± 1.16	0.05 ± 0.03	2.55 ± 0.02
7		EA	0.33 ± 0.03	0.11 ± 0.01	22.94 ± 0.81	0.05 ± 0.01	0.51 ± 0.01
8		AQ	8.62 ± 2.88	10.97 ± 3.06	11.64 ± 0.08	5.51 ± 1.15	1.64 ± 0.06
9	*S. lomentarius*	EE	17.03 ± 1.60	7.63 ± 2.17	13.24 ± 0.28	0.57 ± 0.07	148.08 ± 23.41
10		Hex	25.35 ± 2.26	5.76 ± 0.98	13.46 ± 0.48	0.26 ± 0.01	21.74 ± 4.40
11		EA	1.91 ± 0.15	2.95 ± 0.41	13.68 ± 0.78	0.26 ± 0.04	2.72 ± 0.04
12		AQ	27.05 ± 2.05	8.78 ± 1.68	13.55 ± 0.66	65.80 ± 12.24	278.08 ± 5.73
13	*S. hemiphyllum*	EE	1.08 ± 0.02	0.24 ± 0.01	17.22 ± 0.28	0.15 ± 0.02	1.73 ± 0.03
14		Hex	0.96 ± 0.03	0.27 ± 0.01	21.18 ± 1.82	0.06 ± 0.05	6.64 ± 0.69
15		EA	0.35 ± 0.01	0.02 ± 0.00	53.74 ± 5.52	0.02 ± 0.00	0.02 ± 0.00
16		AQ	0.52 ± 0.01	0.57 ± 0.00	24.01 ± 1.64	152.61 ± 17.37	8.53 ± 4.34
17	*S. fusiforme*	EE	1.08 ± 0.03	0.50 ± 0.03	18.89 ± 0.20	0.38 ± 0.03	22.00 ± 1.32
18		Hex	1.49 ± 0.02	1.06 ± 0.02	19.25 ± 1.17	0.61 ± 0.02	104.31 ± 23.51
19		EA	0.94 ± 0.02	0.32 ± 0.01	19.47 ± 0.63	0.19 ± 0.02	3.49 ± 0.15
20		AQ	1.01 ± 0.07	0.55 ± 0.05	18.80 ± 0.81	0.22 ± 0.03	5.18 ± 0.18
21	*L. japonica*	EE	1.85 ± 0.06	0.47 ± 0.02	14.35 ± 1.54	0.27 ± 0.01	1.15 ± 0.03
22		Hex	6.37 ± 0.56	1.82 ± 0.03	13.99 ± 1.04	0.31 ± 0.01	0.46 ± 0.01
23		EA	7.02 ± 1.01	21.39 ± 3.67	13.77 ± 0.28	0.18 ± 0.03	3.36 ± 0.15
24		AQ	2.56 ± 0.08	0.43 ± 0.04	14.93 ± 0.61	1.62 ± 0.26	6.69 ± 0.76
25	Vc		0.01 ± 0.00	0.27 ± 0.03	/	/	/
26	Acarbose		/	/	/	0.63 ± 0.06	2.51 ± 0.11

**Table 3 foods-12-03000-t003:** Mass spectrometric data of phlorotannins in the ethyl acetate fraction of six brown algae determined using Q-Exactive HF-X Mass Spectrometry.

No.	Rt (min)	Precursor Ion MS ^1^	Product Ions MS ^2^	Tentative Assignment	Species	Ref.
		[M-H]^−^, *m/z*	[M-H]^−^, *m/z*			
1	0.61	317	299.05, 187.03	Phlorotannin derivative	LJ	[44]
2	0.66	387	261.89	Phlorethohydroxycarmalol	SH	[10]
3	0.66	267	249.04, 223.04, 221.02	Phlorotannin derivative	SP, LJ, SF	[19]
4	1.01	263	245.89, 111.02	Phlorethohydroxycarmalol	SL	[10]
5	1.28	317	272.88	Phlorotannin derivative	SL	[44]
6	1.30	385	341.01, 312.90, 261.06, 245.04	Phlorotannin derivative	LJ, SL	[44]
7	1.36	317	298.84	Phlorotannin derivative	SP	[44]
8	1.47	247	228.96, 202.92, 166.83	Dibenzodioxine-1,3,6,8-tetraol	SP, LJ, UP, SF	[19]
9	1.69	317	187.04	Phlorotannin derivative	SH	[44]
10	1.93	361	317.03, 298.86, 273.04	Phlorotannin derivative	SH	[44]
11	2.18	373	247.02, 229.01	Trifucol/fucophlorethol	SH	[10]
12	3.28	621	461.17	Fucotriphlorethol	SH, LJ	[19]
13	3.92	317	298.92, 228.89	Phlorotannin derivative	SF, UP	[44]
14	4.06	265	247.09, 221.12, 193.12	Bifuhalol	SL, SF	[19]
15	4.18	267	222.96	Phlorotannin derivative	SH	[19]
16	4.37	263	245.12, 219.14	Phlorethohydroxycarmalol	SL, SF	[45]
17	4.60	317	298.92, 228.89	Phlorotannin derivative	SH	[44]
18	4.65	369	325.16	Dioxinodehydroeckol	SF	[10]
19	4.67	395	351.18	Phlorotannin derivative	SL	[19]
20	4.95	361	317.03, 298.85, 273.04	Phlorotannin derivative	SF	[44]
21	4.96	369	351.22	Dioxinodehydroeckol	SP	[10]
22	5.06	267	222.96, 220.85	Phlorotannin derivative	UP	[19]
23	5.13	385	261.22	Phlorotannin derivative	SL	[44]
24	5.13	395	351.18, 249.11	Phlorotannin derivative	SF	[19]
25	5.24	369	351.18	Dioxinodehydroeckol	SL	[10]
26	5.24	373	329.21	Fucophlorethol	SL	[10]
27	5.29	421	403.18, 377.17, 213.16	Phlorotannin derivative	SL	[19]
28	5.46	387	329.23	Phlorethohydroxycarmalol	SL, SF	[10]
29	5.56	267	223.13, 195.10, 177.09	Phlorotannin derivative	SL	[19]
30	5.79	317	299.16	Phlorotannin derivative	SL, SF	[44]
31	5.90	385	367.21, 260.93	Phlorotannin derivative	SF	[44]
32	5.90	421	377.19	Phlorotannin derivative	SF	[19]
33	6.14	369	351.22	Dioxinodehydroeckol	SF	[10]
34	6.24	287	269.15, 243.18	Phlorotannin derivative	SL	[19]
35	6.24	317	272.92, 228.93, 187.36	Phlorotannin derivative	SP, SH	[44]
36	6.27	373	329.21, 305.18, 126.90	Triphlorethol/trifucol/fucophlorethol	UP, SL	[10]
37	6.37	363	318.91, 274.92	Phlorotannin derivative	SF	[19]
38	6.43	363	318.91, 274.92	Phlorotannin derivative	SP	[44]
39	6.45	369	351.22	Dioxinodehydroeckol	SL	[10]
40	6.48	267	249.19, 222.96	Phlorotannin derivative	SF	[19]
41	6.54	469	425.25, 264.94	Phlorotannin derivative	SF	[19]
42	6.71	361	298.86	Phlorotannin derivative	SH	[44]
43	6.76	403	259.09	Phlorotannin derivative	SF	[44]
44	6.79	287	269.21, 243.17, 214.93, 172.92	Phlorotannin derivative	SF, SP	[19]
45	6.94	265	247.17, 221.01, 177.02, 168.78	Bifuhalol	SF, SP, SL	[19]
46	6.95	317	299.20, 273.22, 255.21, 245.19	Phlorotannin derivative	SL	[44]
47	6.96	479	443.24, 433.19, 206.90	Fucofurodiphlorethol	SP, SL, SF	[44]
48	7.36	711	675.36, 371.33	Phlorotannin derivative	SF	[44]
49	7.39	267	249.19, 222.96, 195.14	Phlorotannin derivative	SP	[19]
50	7.75	317	299.16, 255.21	Phlorotannin derivative	SL	[44]
51	7.79	371	335.22, 317.21	Eckol	SL	[10]
52	7.94	317	299.20, 273.22, 255.21	Phlorotannin derivative	SP, SF	[44]
53	8.00	509	372.89, 304.91	Phlorotannin derivative	SF	[10]
54	8.05	385	219.14	Phlorotannin derivative	SL	[44]
55	8.15	267	249.19, 223.21	Phlorotannin derivative	SF, SL	[19]
56	8.24	689	653.37, 552.50	Phlorotannin derivative	SF	[44]
57	8.40	363	317.21	Phlorotannin derivative	SP, SF	[19]
58	9.11	711	693.47, 267.20, 249.19	Phlorotannin derivative	SH	[44]
59	10.81	555	164.99	Phlorotannin derivative	SL	[44]

UP: *U. pinnatifida*, SP: *S. pallidum*, SL: *S. lomentarius*, SH: *S. hemiphyllum*, SF: *S. fusiforme*, LJ: *L. japonica*.

## Data Availability

The data that support the findings of this work are available from the corresponding author upon reasonable request.

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
