# Peer review of "Investigation of the Potential Phlorotannins and Mechanism of Six Brown Algae in Treating Type II Diabetes Mellitus Based on Biological Activity, UPLC-QE-MS/MS, and Network Pharmacology"

_foods, 2023, doi:10.3390/foods12163000_

Round 1
Reviewer 1 Report
Dear Author, I reviewed the manuscript (foods-2484355) entitled Investigation of the potential phlorotannins and mechanism of 6 brown algae in treating type II diabetes mellitus based on biological activity, UPLC-QE-MS/MS, and network pharmacology. This manuscript presents relevant information about phlorotannin's antioxidant properties. However, some sections of the presented data can be improved. For this reason, I consider that this manuscript needs minor changes to be considered .
Additional comments.
Highlight the advantages of using phlorotannins as functional foods.
Check paragraphs extension in this manuscript.
Include an experimental design that contains statistical factors and variables of response in the statistical analyses applied to the findings of this research.
Compare the obtained findings with similar assays where other algae or phlorotannins-like compounds have antioxidant modes of action in treating diabetes.
Include future trends to keep working with the obtained data.
Try to conclude with a general statement of the most relevant part of this study.
Author Response
Dear Reviewer:
Journal: Foods
Manuscript Number: foods-2484355
Title: Investigation of the potential phlorotannins and mechanism of 6 brown algae in treating type II diabetes mellitus based on biological activity, UPLC-QE-MS/MS, and network pharmacology
First of all, thanks very much for your kind consideration of our manuscript.
Those comments are valuable and very helpful for revising and improving our paper, as well as the important guiding significance to our research. According to the comments from reviewer, we made all corrections and provided relevant explanations about request raised by the reviewer; the corresponding changes were highlighted with yellow color in the present revised form.
Here, we submit the revised manuscript. We hope this revised manuscript will be reported in Foods. We appreciate so much for your kind consideration.
Sincerely yours,
Dr.Yan Ding, E-mail: dingyan_515@hotmail.com
School of Food Science and Technology, Dalian Polytechnic University
Dalian, Liaoning 116034, PR China.
Tel. +86-411-8632-3656, Fax. +86-411-8632-3656.
Answers for Comments of Reviewer
We appreciate so much your kind consideration of our manuscript. According to reviewer's opinions, some parts of the manuscript have been corrected. Revised portions are highlighted with yellow color on the paper. The main corrections in the paper and the responses to the reviewer's comments are listed as follows:
Dear Author, I reviewed the manuscript (foods-2484355) entitled Investigation of the potential phlorotannins and mechanism of 6 brown algae in treating type II diabetes mellitus based on biological activity, UPLC-QE-MS/MS, and network pharmacology. This manuscript presents relevant information about phlorotannin's antioxidant properties. However, some sections of the presented data can be improved. For this reason, I consider that this manuscript needs minor changes to be considered.
Response: Thank you so much for your positive feedback and great patience. We have revised our manuscript according to your comments and suggestions.
Comment#1: Highlight the advantages of using phlorotannins as functional foods.
Response: Thank you very much for your questions about this manuscript. The advantages of phlorotannins for use as functional foods are highlighted in the Introduction to the manuscript. The additions are listed below and are marked with a yellow background:
Phenolic compounds, the most studied phytochemicals, are ubiquitous and abundant in all plant-based diets (in line 59-60).
As polyphenols unique to brown algae, phlorotannins are different from terrestrial plant polyphenols consisting of gallic acid and flavonoid polymers (in line 62-63).
Some studies have even reported that there is great potential for the use of phlorotannins as a functional ingredient in the development of new and improved food products, mainly in the areas of dairy and seafood products (in line 70-72).
Comment#2: Check paragraphs extension in this manuscript. Include an experimental design that contains statistical factors and variables of response in the statistical analyses applied to the findings of this research.
Response: Thank you. We have recheck the paragraph extension, and adjusted the paragraph assignments in this manuscript.
Comment#3: Compare the obtained findings with similar assays where other algae or phlorotannins-like compounds have antioxidant modes of action in treating diabetes. Include future trends to keep working with the obtained data.
Response: Thank you for your conscientiousness and responsibility. The results obtained be compared with other similar experiments in this manuscript. The future trends of phlorotannins against T2DM are outlined in the conclusion section of the manuscript. The details are as follows and are marked in the paper with a yellow background.
The results obtained presented similar results with the antioxidant potential of different extract fractions of Sargassum vulgare. Among them, considerable antioxidant potential was shown in the EtOAc fraction, which was more promising than the crude extract, Hex and AQ [19] (in line 273-276).
Considering the synergistic effects of other active substances, further studies combining multiple analytical methods are needed to reveal the active substances in brown algae more comprehensively and accurately. The in vivo and vitro studies supplemented the targets and mechanisms of the effects of phlorotannins on T2DM to promote the development of related new drugs, which provided a certain theoretical basis for the exploitation of the active value of phlorotannins and maximised the potential of brown alga (in line 474-480).
Comment#4: Try to conclude with a general statement of the most relevant part of this study.
Response: A summary statement of the most relevant parts of this study is provided in the conclusion of this manuscript. The details are as follows and are marked in the paper with a yellow background.
The present study was designed to provide a scientific basis for the analysis of potentially active phlorotannins against T2DM by component isolation, bioactivity determination, UPLC-QE-MS/MS, and network pharmacology. Antioxidant activity and inhibition of carbohydrate hydrolases were assessed in 24 groups of brown algae samples, and the structures of phlorotannins and their derivatives were deduced by UPLC-QE-MS/MS. Predicting the mechanism of phlorotannins against T2DM using a network pharmacology approach (in line 451-457).
For Editor:
Comment#1: We found that the plagiarism rate of your paper is high. Based on the publication rules, for new submissions, repetition on the whole can not be higher than 30%, and repetition with one single part must be lower than 6%. Please revise the duplication based on the check report, especially the parts highlighted in red, and make sure that there is no large section of repetition in the published paper.
Response: Thank you. We apologize for the inappropriate content. We modify the duplicate information of the manuscript according to your suggestions: According to your opinion, we have reduced the repetition rate of the manuscript to less than 30%, and for the reference part of each document ≤ 6%.
Comment#2: We have added a new author to the manuscript. This author was primarily responsible for providing the data analysis software and teaching the analysis methods in this study. However, this was overlooked during the writing of the paper, so this author was added in this revision of the paper. New additions are marked in the paper with yellow background (in line 6, 10, 485).

Reviewer 2 Report
This study elucidated the chemical and functional properties of brown algae extracts, such as phenolic content, antioxidative activities, enzyme inhibition activities, component identification, and computational screening for T2DM treatment. Brown algae abundant in phlorotannins seems an intriguing and important natural source for treating T2DM. You conducted enthusiastic research with various technical methods. However, I would recommend you consider the following points.
l Title
The word mechanism in the title is not sure. What kind of mechanism was described in the paper?
l Mass analysis
- The assignment should be with references in Table 3.
- [M-H]+ is shown in Figure 3, however, [M-H]- is described in the text. Which is correct?
- Fragmentation of benzene ring such as m/z44 in Figure 3 is questionable for me. Please cite the reference.
l Figure 6 and 7
Figure 6 is too large and Figure 7 is too small to read the text.
l Conclusions
I feel the conclusion is a little exaggeration because the function related to T2DM is in silico.
Author Response
Dear Reviewer:
Journal: Foods
Manuscript Number: foods-2484355
Title: Investigation of the potential phlorotannins and mechanism of 6 brown algae in treating type II diabetes mellitus based on biological activity, UPLC-QE-MS/MS, and network pharmacology
First of all, thanks very much for your kind consideration of our manuscript.
Those comments are valuable and very helpful for revising and improving our paper, as well as the important guiding significance to our research. According to the comments from reviewer, we made all corrections and provided relevant explanations about request raised by the reviewer; the corresponding changes were highlighted with yellow color in the present revised form.
Here, we submit the revised manuscript. We hope this revised manuscript will be reported in Foods. We appreciate so much for your kind consideration.
Sincerely yours,
Dr.Yan Ding, E-mail: dingyan_515@hotmail.com
School of Food Science and Technology, Dalian Polytechnic University
Dalian, Liaoning 116034, PR China.
Tel. +86-411-8632-3656, Fax. +86-411-8632-3656.
Answers for Comments of Reviewer
We appreciate so much your kind consideration of our manuscript. According to reviewer's opinions, some parts of the manuscript have been corrected. Revised portions are highlighted with yellow color on the paper. The main corrections in the paper and the responses to the reviewer's comments are listed as follows:
This study elucidated the chemical and functional properties of brown algae extracts, such as phenolic content, antioxidative activities, enzyme inhibition activities, component identification, and computational screening for T2DM treatment. Brown algae abundant in phlorotannins seems an intriguing and important natural source for treating T2DM. You conducted enthusiastic research with various technical methods. However, I would recommend you consider the following points.
Response: Thank you so much for your positive feedback and great patience. We have revised our manuscript according to your comments and suggestions.
Comment#1: In Title, the word mechanism in the title is not sure. What kind of mechanism was described in the paper?
Response: Thank you very much for your questions regarding this manuscript. Predicting the mechanism of phlorotannins against T2DM using network pharmacology approach. A total of 43 T2DM-related targets were obtained by network pharmacological prediction, mainly focusing on EGFR tyrosine kinase inhibitor resistance, endocrine resistance, PI3K-Akt signaling pathway, and Rap1 signaling pathway. Based on this, the word mechanism is used in the title.
Comment#2: In Mass Analysis, the assignment should be with references in Table 3.
Response: Thank you, the references of mass spectrometry data of phlorotannins in Table 3 has been annotated.
Comment#3: In Mass Analysis, [M-H]+ is shown in Figure 3, however, [M-H]- is described in the text. Which is correct?
Response: Thank you very much for your comments, in the mass analysis, [M-H]- is correct and the wrong part of the text has been corrected.
Comment#4: In Mass Analysis, Fragmentation of benzene ring such as m/z44 in Figure 3 is questionable for me. Please cite the reference.
Response: Thanks to your seriousness, we verify the fragmentation m/z44 of the benzene ring in Fig. 3 by citing the relevant literature.
[10] Erpel, F.; Mateos, R.; Pérez-Jiménez, J.; Pérez-Correa, J.R. Phlorotannins: From isolation and structural characterization, to the evaluation of their antidiabetic and anticancer potential. Food Res Int 2020, 137, 109589.
[19] Chouh, A.; Nouadri, T.; Catarino, M.D.; Silva, A.M.S.; Cardoso, S.M. Phlorotannins of the Brown Algae Sargassum vulgare from the Mediterranean Sea Coast. Antioxidants 2022, 11(6), 1055.
[44] Catarino, M.D.; Silva, A.M.S.; Mateus, N.; Cardoso, S.M. Optimization of Phlorotannins Extraction from Fucus vesiculosus and Evaluation of Their Potential to Prevent Metabolic Disorders. Mar Drugs 2019, 17(3), 162.
Comment#5: Figure 6 is too large and Figure 7 is too small to read the text.
Response: Thank you very much for your valuable comments. The dimensions of figures in the manuscript have been appropriately modified.
Comment#6: I feel the conclusion is a little exaggeration because the function related to T2DM is in silico.
Response: Thank you very much for your valuable comments on this manuscript. We have adjusted some parts of the conclusion. The details are as follows:
A total of 43 T2DM-related targets were obtained by network pharmacological prediction, mainly focusing on EGFR tyrosine kinase inhibitor resistance, endocrine resistance, PI3K-Akt signaling pathway, and Rap1 signaling pathway (in line 470-473).
For Editor:
Comment#1: We found that the plagiarism rate of your paper is high. Based on the publication rules, for new submissions, repetition on the whole can not be higher than 30%, and repetition with one single part must be lower than 6%. Please revise the duplication based on the check report, especially the parts highlighted in red, and make sure that there is no large section of repetition in the published paper.
Response: Thank you. We apologize for the inappropriate content. We modify the duplicate information of the manuscript according to your suggestions: According to your opinion, we have reduced the repetition rate of the manuscript to less than 30%, and for the reference part of each document ≤ 6%.
Comment#2: We have added a new author to the manuscript. This author was primarily responsible for providing the data analysis software and teaching the analysis methods in this study. However, this was overlooked during the writing of the paper, so this author was added in this revision of the paper. New additions are marked in the paper with yellow background (in line 6, 10, 485).

Round 2
Reviewer 2 Report
I think the revised manuscript is very carefully and nicely revised by responding to the reviewer's concerns.